# Preferred music-listening level in musicians and non-musicians

**Antonia Olivia Dolan, Emanuele Perugia, Karolina Kluk***

Manchester Centre for Audiology and Deafness (ManCAD), School of Health Science, The University of Manchester, Manchester, United Kingdom

* karolina.kluk@manchester.ac.uk

## Abstract

The purpose of this study was to establish whether preferred music-listening level differed between musicians and non-musicians, and whether preferred music-listening level was related to music genre preference and lifetime noise exposure. Seventeen musicians (mean age = 29.06 years, SD = 4.74; female n = 9) and 17 non-musicians (mean age = 28.94 years, SD = 4.63; female n = 9) with clinically normal hearing were recruited to listen to six music samples from different genres and one sample of environmental sounds. Participants adjusted the listening level [dB(A)] until the music was loud and enjoyable. This was repeated three times and an average was taken. Lifetime noise exposure was estimated using the Noise Exposure Structured Interview. Preferred music-listening levels of musicians were significantly higher than non-musicians. The preferred music-listening level differed with genre preference, with the participants' favorite tracks being played at 11 dB higher level than the least favorite tracks. There was also a positive correlation between lifetime noise exposure and preferred music-listening level. Musicians prefer to listen to music at higher level than non-musicians and thus may be more susceptible to noise induced hearing loss than non-musicians. As such, musicians in particular would benefit from simple changes in lifestyle and listening habits, including increased awareness of the risks of higher listening levels, as well as the use of hearing protection.

**Data Availability Statement:** All relevant data are within the paper and its Supporting Information files.

**Funding:** * AOD, EP, KK NIHR (National Institute for Health Research) Manchester Biomedical Research Centre * EP, KK MRC (Medical Research

## Introduction

Noise induced hearing loss (NIHL) is the second largest cause of hearing loss in the UK [1], with presbycusis being the first. NIHL can be caused by occupational/industrial noise [2] or environmental sounds [3]. In addition, NIHL may also be caused by listening to music at excessively high sound levels [4].

There is a considerable literature on the perspective of musicians in relation to NIHL and hearing protection. These studies revealed that whilst musicians are aware of the adverse effects that loud music listening can have on their hearing, they are more concerned with the satisfaction obtained from this and the inferior sound quality that they perceive is associated with the use of hearing protection [5–9]. This suggests that listening to music at lower/quieter sound levels decreases the level of enjoyment and/or satisfaction that may be achieved from listening at higher/louder sound levels [10, 11].

Council), UK (MR/L003589/1) The funders had no role in study design, data collection and analysis, decision to publish, or preparation of the manuscript.

**Competing interests:** The authors have declared that no competing interests exist.

The underlying mechanisms for individuals choosing to listen to music at excessively high levels has been a topic of research over the years. Research by Todd & Cody [11] and Todd [12] suggests that this may be related to the acoustic sensitivity of the vestibular system in the 100 to 500 Hz range. Todd [12] monitored music listening and activation of the vestibular system and found that vestibular activation can be evoked by loud music presented at a level above 90 dB SPL since it contains considerable energy at low frequencies (below 500 Hz). Todd argued that when vestibular activation occurs, stimulation follows pathways that reach the "reward centers" of the brain; formally referred to as the limbic system. This stimulation is referred to as a 'hedonic response' and may provide an explanation as to why loud stimulus is enjoyable to individuals. Other research in this area suggests that emotional involvement [13] and behavioral characteristics such as addiction [14, 15] may also contribute to why individuals enjoy listening to loud music.

It has previously been observed that musical (genre) preference has some effect on loudness perception. Fucci [16] used magnitude-estimation scaling (1–7) to record how loud an individual perceived a piece of rock music. The study revealed that the group of participants who preferred rock music provided lower numerical responses, compared to the group of participants who stated they disliked rock music. This indicates that when music is enjoyed, it is perceived to be quieter. However, a limitation of this work is that it was based on a subjective self-report measure and did not provide information about preferred loudness.

Thus far, there are no reliable studies, with adequate sample sizes and group controls, which assess whether preferred music-listening level differs between musicians and non-musicians, using an objective measure of sound level. Previous research in this area is limited to the work of Hoover & Cullari [17], in which they attempted to compare subjective perception of loudness and musical preference in musicians and non-musicians. This was investigated using a loudness matching task. Participants were asked to rate music samples on a 1 to 7 Likert scale; 1 meaning least preferred and 7 most preferred. They were then asked to match 10 music samples of various genres, to a neutral stimulus (pink noise). However, Hoover & Cullari [17] did not report the preferred music listening levels, but only the accuracy of musicians and non-musicians to match the level of the music samples to the level of pink noise.

Therefore, the present study aimed to investigate whether preferred music-listening level is different between musicians and non-musicians, and whether musical preference influences preferred music-listening level. As the results from these investigations may provide evidence that one group (musicians or non-musicians) is exposing themselves to greater amounts of noise, the present study will also assess whether lifetime noise exposure correlates with preferred music-listening level.

The current study aimed to: 1) establish whether preferred music-listening level is different between musicians and non-musicians; 2) assess whether there is a relation between music preference and preferred music-listening level; 3) assess whether lifetime noise exposure correlates with preferred music-listening level.

## Materials and methods

### Participants

A total of 34 participants, 17 musicians and 17 non-musicians, were recruited. The two groups were matched in gender and age (musicians consisted of eight males and nine females; mean age = 29.06 years; Standard Deviation (SD) = 4.74 and non-musicians consisted of eight males and nine females; mean age = 28.94; SD = 4.63). Participants were recruited using flyers advertising the research, around the University of Manchester campus and at the 'Futureworks' campus (i.e., higher education setting for music performance and audio engineering). The

musicians ranged from guitarist/bassist, drummer to saxophonist. They played in rock and jazz bands.

All participants provided written consent. The study was approved by the University of Manchester Research Ethics Committee (review reference: 2019-6217-10064).

## Inclusion criteria

The inclusion criteria for musicians were: aged between 20–40 and regularly rehearsing their instrument or playing at gigs/concerts (self-reported). The inclusion criteria for non-musicians were: aged between 20–40 and no experience playing a musical instrument. All participants were required to have clinically normal hearing thresholds ($\leq$ 20 dB HL at 250–8000 Hz), normal tympanometry values according to the British Society of Audiology [18], i.e. Ear canal volume: 0.6–2.5 cm$^3$; Compliance: 0.3–1.6 cm$^3$ and Pressure: -50 - + 50 daPa) and clear otoscopy.

## Equipment

**Hearing assessment.**  Pure-tone audiometry was carried out using a GSI Pello Audiometer in a double-walled sound-attenuating booth. Tympanometry was carried out using a GSI Tympstar Pro.

**Preferred music-listening level test.**  A Denon DCD685 CD Player was used to play the chosen music samples and was routed to a GSI61 Audiometer using line output connections to the speech inputs on the GSI61 Audiometer. The system was calibrated with a B&K2250 sound level meter and a B&K4153 artificial ear, in order to ensure that the audiometer dial and the output dB SPL were consistent (within 3 dB). Participants used a pair of Sennheiser HD650 Headphones to listen to the music samples and adjusted the sound level dial on the audiometer.

To ensure that the overall sound level of all music samples was similar, the seven auditory stimuli samples used were normalized using a digital audio workshop (Pro Tools). The data displayed in S1 Table show the respective sound levels that correspond to the audiometer display of 70, 80, 90, and 100 dB HL. All values are reported in dB(A), measured using the B&K 2250 sound level meter with a free field half inch pre-polarized microphone (type 4139) and the ZC0032 microphone pre-amplifier. The readings were done over a 60 second interval, with the analysis window centered at 500 Hz (one octave wide).

## Procedures

All participants performed the experiment in a single session lasting approximately 30 minutes.

**Hearing assessment (otoscopy, tympanometry and pure tone audiometry).**  Each test session commenced with a full clinical hearing assessment which included otoscopy, tympanometry, and pure-tone audiometry; all procedures followed guidelines published by the British Society of Audiology [18, 19]. Otoscopy and tympanometry were used to ensure that any participants with outer or middle ear abnormalities (i.e., tympanic membrane perforation, active infection, occluded wax, or foreign bodies) were prohibited from progressing further into the experiment.

Pure-tone audiometry was performed in each ear, to obtain hearing thresholds between 250 Hz and 8000 Hz as per BSA guidance [19]. The inclusion criteria were hearing thresholds $\leq$ 20 dB HL in both ears at all frequencies tested (250–8000 Hz). This was to ensure only participants with hearing thresholds considered 'normal' ($\leq$ 20 dB HL) were recruited as hearing

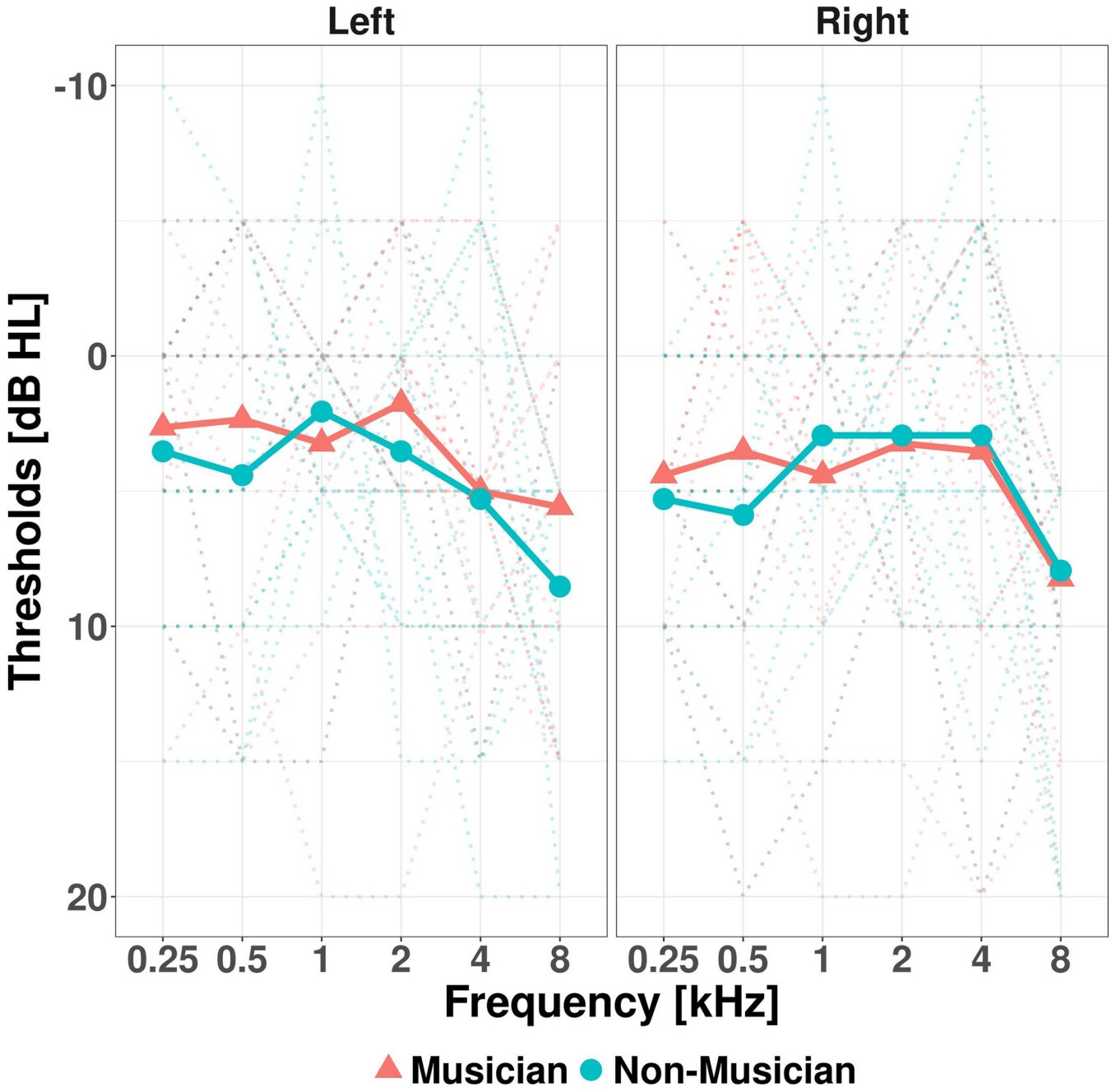

**Fig 1. Individual (dotted lines) and mean (solid lines) pure tone audiometry results.**

loss could affect their preferred music-listening level. Fig 1 shows a summary of all hearing thresholds.

Using the average over the two ears, musicians and non-musicians had similar average thresholds from 250 to 8000 Hz ($W = 4669.5$, $p = 0.202$), musicians had the median = 2.5 dB HL (SD = 5.931), whereas the non-musicians had median = 5 dB HL (SD = 6.151).

**Preferred music-listening level.** Following the clinical hearing assessment, participants were presented with six music samples and one environmental sound, which they listened to

using over-ear headphones (Sennheiser HD650) in a quiet testing room. The music/environmental samples lasted 60 seconds and varied in genre to assess relation between music preference and preferred music-listening level. The six music samples were: 1) Whole Lotta Love by Led Zeppelin, 2) Heartbeats by Jose Gonzales, 2) Crazy In Love by Beyonce, 3) Sad But True by Metallica, 4) Virtual Insanity by Jamiroquai, 6) Symphony No. 5 in C Minor, Op. 67: I. Allegro con brio by Ludwig Van Beethoven et al. Whilst, the environment sound was bird song, which was a sample taken from a high-quality recording of birds signing. Spectral analysis of the music samples selected were carried out to ensure they contained frequency components around 500 Hz. This was an important aspect of the study, as Todd & Cody [11] and Todd [12] showed that 100–500 Hz is the frequency region that the vestibular system is most sensitive to. These authors suggested that this otolithic sensitivity may play some role in satisfaction obtained from loud music listening.

The participants were asked to adjust a dial controlling sound level (in 2-dB increments) until each music sample was suitably loud and enjoyable. This process was repeated three times and an average value was determined for each music sample. The repetitions were performed consecutively. The participants were asked to state their favorite and least favorite music sample.

In order to protect the participants' hearing, personal exposure level calculations were carried out, which established that participants should not breach 95 dB HL read out on the audiometer [equates to approximately 100 dB(A); see S1 Table]. The researcher supervised each participant and ensured that they did not breach this level, by stopping the CD player (to stop the music) if and when necessary. For safety reasons, participants who were listening to music at loud levels [>70 dB (A)] were advised to limit their noise exposure for the next 24 hours.

**Noise exposure.** The lifetime noise exposure of each participant was then estimated using the Noise Exposure Structure Interview (NESI) developed by Guest et al. [20]. The interview focuses on areas of the interviewee's life in which they were exposed to high levels of noise. The areas are split into three sections: recreational, occupational, and firearm use. In each area, participants were asked to state periods of their life where they may have been in 'noisy' situations. Guest et al. [20] stated that an environment can be described as 'noisy', if two normal hearing individuals need to raise their voice to communicate when they are at a distance of 4 feet or 1.2 m [equivalent to 87 dB(A) and above].

The exposure duration was then estimated by asking the interviewee to average the time spent in each environment, using years, weeks, days, and hours. To estimate the exposure level, participants were asked to refer back to the scenario of communicating with an individual approximately 4-feet away and state the level of vocal effort required (i.e., talk normally, raise voice, shout, or move closer). In the interviews, the supplementary materials provided by [20], such as guidance (with examples and tables) and spreadsheet, to calculate the total units of lifetime noise exposure was used. The interviews took about 20 minutes to complete. The resulting lifetime noise exposure score is linearly related to the total energy of noise exposure above 80 dB(A). That is, one unit is equivalent to one working year (2080 hours) of exposure to 90 dB(A). The NESI also records details relating to hearing protection use, which is used to adjust the estimate of lifetime noise exposure. For a given activity, the cumulative units of noise exposure are calculated as:

$$U = \frac{Y * W * D * H}{2080} * \left[ P * 10^{\frac{L-A-90}{10}} + (1 - p) * 10^{\frac{L-90}{10}} \right]$$

Where $U$ is units of noise exposure (in linear units), $Y$ is years of exposure, $W$ is weeks per year of exposure, $D$ is days per week of exposure, $H$ is hours per day of exposure, *2080* is the number of hours in a working year, $L$ is the estimated sound level in dBA, $A$ is the attenuation

of hearing protection (dB), and *P* is the proportion of time that hearing protection was worn (0 to 1). The values of *U* were calculated for each activity and then summed.

**Analysis and statistical methods.** Statistical analyses were completed in R (version 3.6.3, [21]). Analysis was focused on assessing differences and correlations. The data were non-normally distributed and as such non-parametric tests were used. To establish differences in preferred music-listening level between the two groups, a Wilcoxon rank-sum test was conducted. To assess the relation between music preference and preferred music-listening level, a Wilcoxon signed-rank test was used. A possible correlation between an individual's estimated noise exposure levels and their preferred music-listening level was investigated using the Spearman's rank correlation coefficient.

In order to untangle the effects of NESI and musicianship on the preferred music-listening level, and since the NESI and musicianship are strongly related to each other, two multiple regression models were run with alpha of 0.05. The dependent variable was always the preferred music-listening level. The predictors in the first model were pure tone average (PTA: average thresholds at octave frequencies from 250 to 8000 Hz of both ears), NESI and musicianship; the second model had only PTA and NESI as predictors. All variables were standardized.

## Results

Fig 2 shows boxplots of the average level of music [in dB(A)] at the preferred music-listening level for each group. The median (reported here since the data were not normally distributed) was higher in musicians [median = 80.61 dB(A) and SD = 6.89 dB] compared to non-musicians [median = 73.61 dB(A) and SD = 7.53 dB], and this difference was significant (W = 241.5, $p < 0.001$) with a Cohen's d of 1.139 (a large effect size).

Participants preferred to listen to their favorite tracks at higher level than their least favorite tracks. Median preferred music-listening level for their favorite tracks was 80.17 dB(A), SD = 8.41 dB. Median preferred music-listening level for their least favorite tracks was 69.50 dB(A), SD = 7.86 dB. The Wilcoxon signed-rank test revealed a significant difference in preferred listening-level between participants' favorite and least favorite tracks (W = 595, $p < 0.001$).

In addition, participants also preferred to listen to the music samples at a higher level than the environmental bird song sound sample. Median preferred listening level for the music samples was 77.22 dB(A), SD = 8.24 dB. Median preferred listening level for the environmental bird song sample was 69.50 dB(A), SD = 9.02 dB. The Wilcoxon signed-rank test was significant (W = 564, $p < 0.001$).

Overall, the NESI scores were between 1.58 and 126.78. On average 81% of the NESI score was driven by recreational activities, whilst the 19% was due to occupational activities. All 34 participants took part in some 'noisy' recreational activities. However, 24 participants, of which 15 were musicians, reported 'noisy' occupational activities. Twelve participants (7 musicians) used firearms but with a decimal NESI score. The hearing protection used ranged from Custom in-the-ear (IE) Plugs (ER Bang series or similar alternatives with filters used according to instrument played) to standard foam ear plugs.

To investigate a possible correlation between an individual's estimated noise exposure levels and their preferred music-listening level, a Spearman's rank correlation coefficient was calculated. A scatterplot of the data can be found in Fig 3. The Spearman's rank correlation coefficient was significant (r = 0.58, $p < 0.001$) showing a moderate positive correlation between estimated lifetime noise exposure and preferred music-listening level.

An additional Wilcoxon's Rank-Sum Test revealed a significant difference in NESI scores (W = 283, $p < 0.001$) between the two groups. Median NESI score for musicians was 52.03,

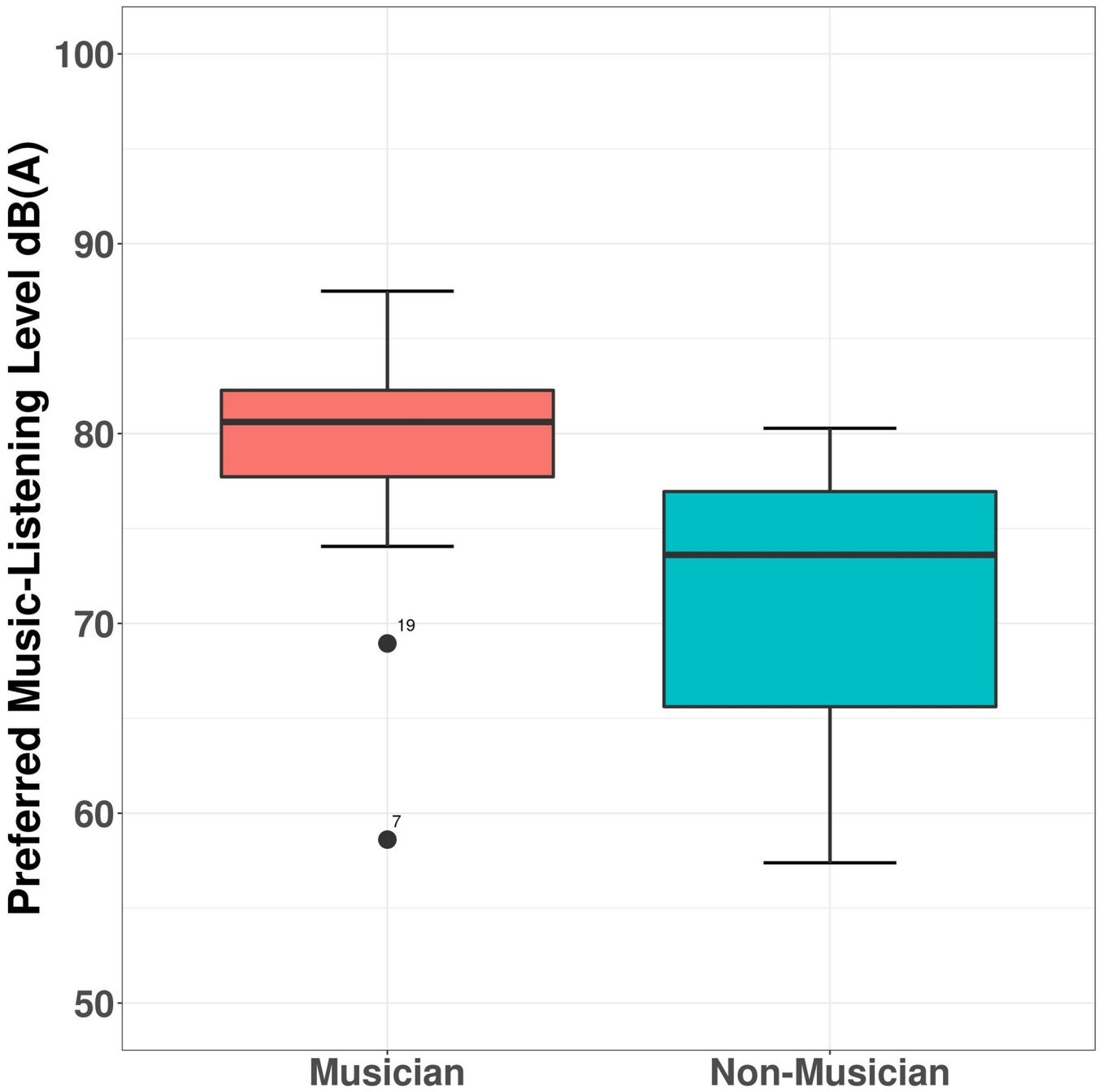

**Fig 2. Preferred music-listening level for musicians and non-musicians.** The boxplots show the smallest observation, 25% quantile, median, 75% quantile and the largest observation.

SD = 22.31. Median NESI score for non-musicians was 6.52, SD = 10.27 with a Cohen's d of 2.747. A boxplot of the data is shown in Fig 4.

## Multiple regression

The model with PTA, NESI and musicianship was significant [Adjusted $R^2$ = 0.213, $F_{(3,30)}$ = 3.982, $p$ = 0.017] with PTA ($\beta$ = -0.043, $p$ = 0.783), NESI ($\beta$ = 0.277, $p$ = 0.309), and

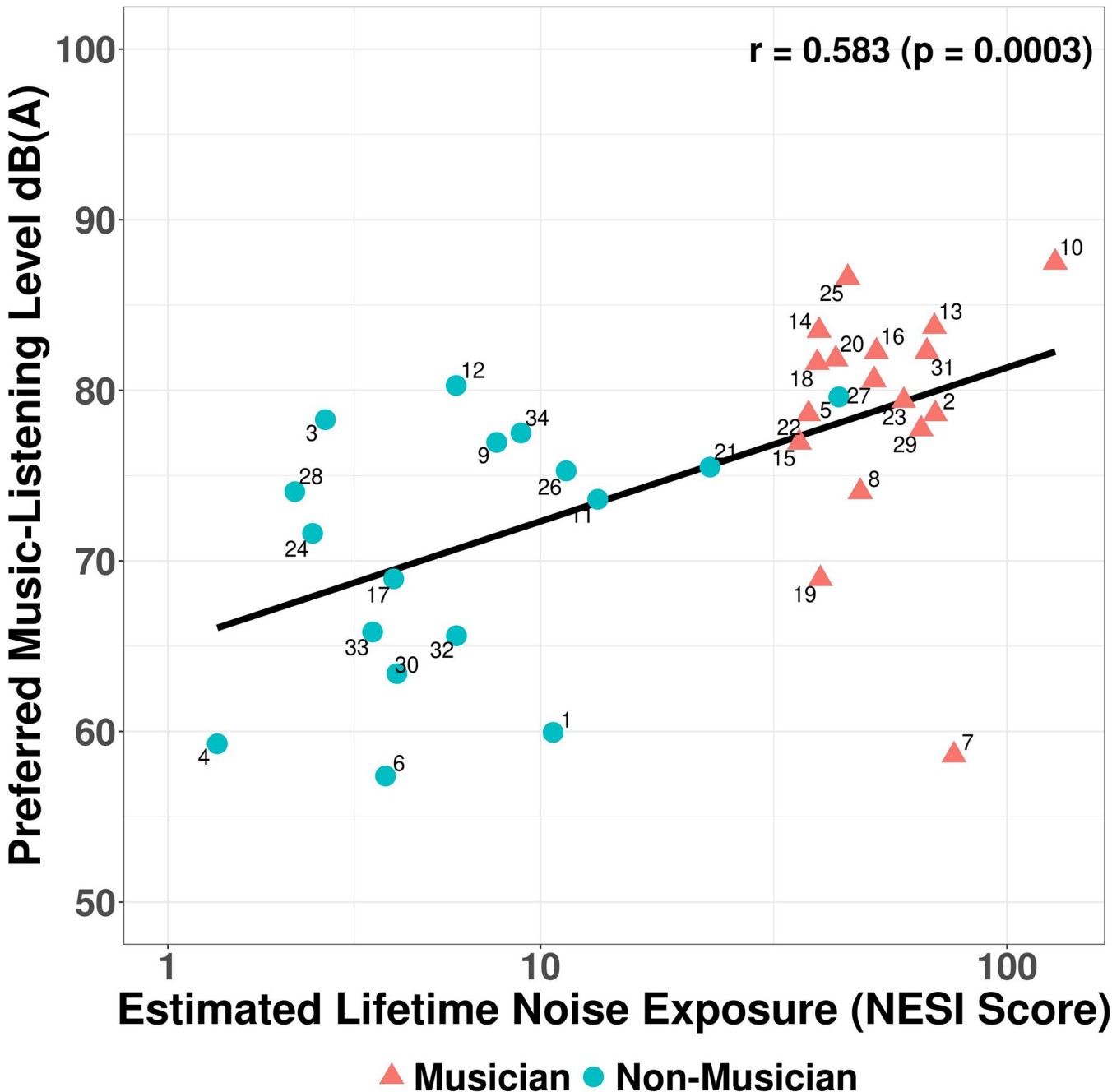

**Fig 3. Preferred music-listening level as a function of lifetime noise exposure (NESI score).**

musicianship (β = -0.544, $p$ = 0.311) being not significant predictors for the preferred music-listening level. The magnitude of the regression coefficient for musicianship was large suggesting that non-musicians had preferred music-listening level lower than musicians, on average. However, this model was unstable due to the collinearity, which lead to increase in the standard error of a regression coefficient, which in turn lead to increase in confidence interval and decrease in the $p$-values of the coefficient [22, 23]. The impact of collinearity is estimated using the variance-inflation factor (VIF) [22, 23]. The VIF for the NESI and musicianship was about

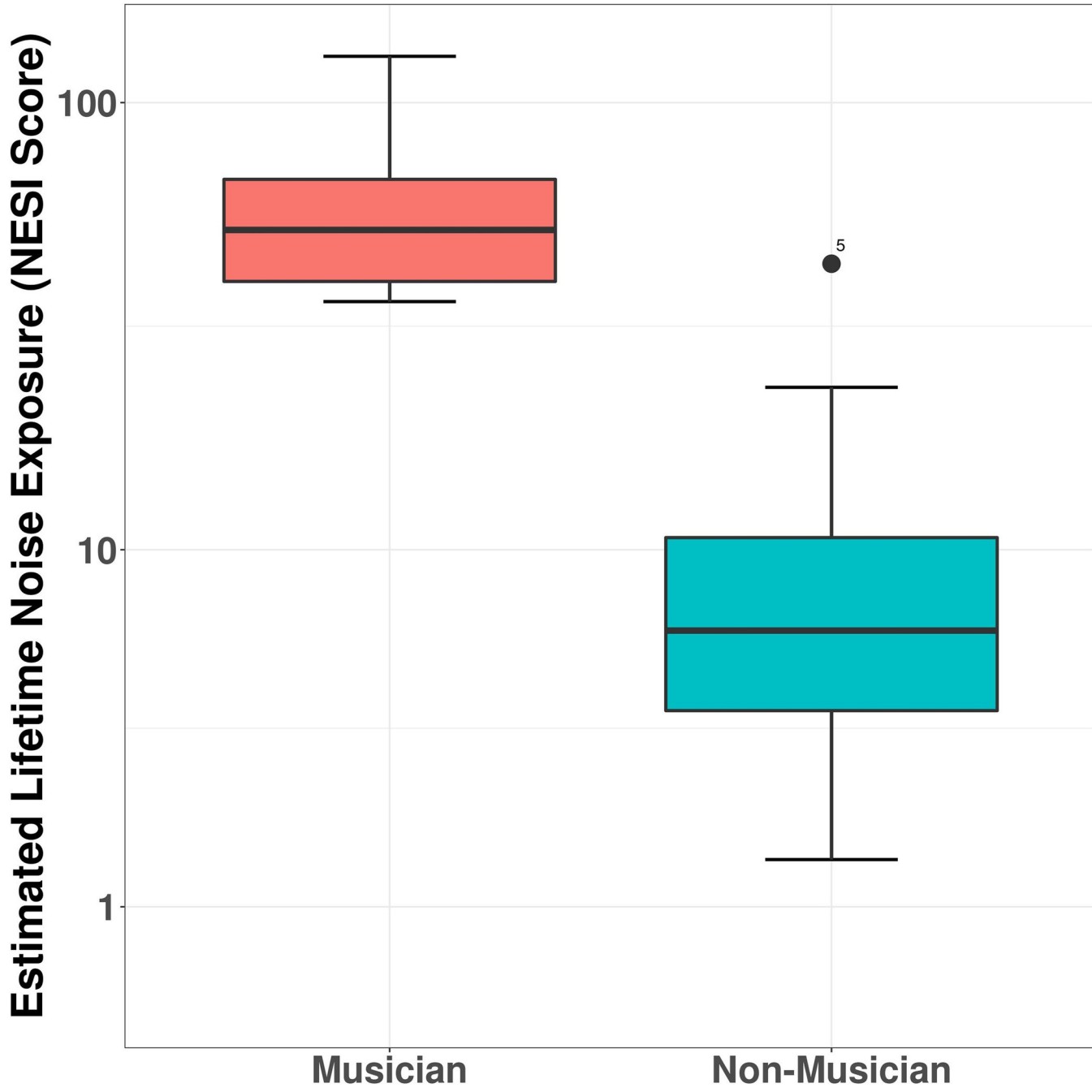

**Fig 4. NESI scores for musicians and non-musicians.** The boxplots show the smallest observation, 25% quantile, median, 75% quantile and the largest observation.

3, which is high according to Fox [22]. The model with only PTA and NESI was significant [Adjusted R2 = 0.212, F(2,31) = 5.429, $p$ = 0.01] with the NESI being the only significant predictor ($\beta$ = 0.503, $p$ = 0.003). The coefficient for PTA did not vary significantly ($\beta$ = -0.042, $p$ = 0.791) from that of the first model, so the musicianship did not confound the association with PTA. Instead, the coefficient for NESI increased (from 0.277 to 0.503) and became statistically significant suggesting that the effect of NESI on the preferred music-listening level was confounded by musicianship.

## Discussion

The first aim of the present study was to establish whether there is a difference in preferred music-listening level between musicians and non-musicians. Overall, musicians preferred to listen to music at a higher sound level [dB (A)] than non-musicians. Thus, the hypothesis that musicians prefer to listen to music at higher sound levels can be accepted. The difference in preferred music-listening level may be the result of an intrinsic effect of other differences found between the two groups such as musicians having a stronger emotional experience when listening to music [24] and/or the differences in brain anatomy in areas that process motor, sensory and cognitive performances [25]. Previous studies into differences between musicians and non-musicians have also revealed that musicians have significantly stronger activation of some brain areas, such as the left inferior parietal lobe (IPL), than non-musicians [24, 26]. The IPL has a key role in sensory and emotional perception [24, 27].

The present study also assessed individual musical preference and its effect on preferred music-listening level, as previous research has indicated that sounds that are enjoyed are perceived as being quieter [16, 28]. Irrespective of group, participants listened to music they preferred at a level approximately 11 dB higher than music they liked less. Furthermore, analysis of the descriptive statistics for the dB(A) level which participants preferred for music compared to the bird song, revealed that participants preferred to enjoy the music at a higher level than the bird song. These data are broadly in line with the findings from Cullari & Semanchick [28] and Fucci [16], who assessed loudness perception and music preferences, and their findings support the hypothesis that when an individual does not perceive a sound as 'noise', such sound is perceived to be quieter and therefore could be listened to at a higher listening-level. Listening to music at high levels could put musicians at greater risk of NIHL, particularly if listening to music above "safe" sound levels [85 dB(A) for more than 8 hours a day; e.g., EU Directive 2003/10/EC].

This leads to the third aim of the current study, which was to assess whether there was a correlation between estimated lifetime noise exposure and preferred music-listening level. A moderate positive correlation was found, demonstrating that individuals with greater noise exposure preferred to listen to music at higher levels. Additionally, statistical analysis also revealed that musicians had greater life-time exposure to noise compared to non-musicians. The recruitment was done using flyers around the campus and at the 'Futureworks' campus, so the greater noise exposure in musicians than non-musicians may be due to the intrinsic musicians' interests and works. For instance, 'noisy' occupational activities were reported by 15 out of 17 musicians but only by 9 out of 17 non-musicians. Since all participants had clinically normal hearing, it could be argued that the desire to listen to music at high levels is not likely to be a result of hearing loss caused by noise exposure. However, it is important to note that hearing thresholds were tested following BSA guidelines [19], which involves only testing at 250, 500, 1000, 2000, 4000 and 8000 Hz. Extended high-frequency audiometry was not performed; thus, it is possible that some of our participants had a degree of hearing loss present at very high frequencies (> 8000 Hz) which may have contributed to their preferred music-listening level. It is also important to note that the variation in hearing thresholds within the normal hearing range (from -10 to 20 dB HL) may have also had some effect, but the average PTA was not a significant predictor of preferred listening level in the regression model. However, we cannot rule out any (sub-clinical) hearing damage in musicians as extended high frequency-audiometry or speech-in-noise tests were not performed. Further studies should include these measures as they may provide further understanding for why musicians prefer to listen to music at higher levels than non-musicians.

Furthermore, it may be possible that noise induced cochlear synaptopathy (NICS), loss of the synapses between inner hair cells and auditory nerve fibers, has some contribution to the correlation between noise exposure and preferred music-listening level. Research indicates that, unlike NIHL, NICS is not associated directly with increased pure-tone audiometry thresholds [29–32]. Instead, NICS may affect aspects of auditory information processing such as intensity and temporal coding and could affect listening in background noise [30, 33]. Therefore, it is plausible that preferred music-listening level is one aspect that might be affected by NICS. Couth et al. [34] investigated the perceptual consequences of NICS in musicians using an extensive test battery. Although musicians reported poorer hearing in noise abilities than non-musicians, Couth et al. [34] did not observe any significant differences between musicians and non-musicians, or between high- and low-noise exposure groups, in any of the tests in their test battery. It is possible that this lack of difference in performance between musicians and non-musicians was due to participants with more musical training being better at temporal processing tasks than the naïve participants [35–37].

Although this study did not directly intend to investigate musicians' usage and acceptability of hearing protection, the results obtained from administering the NESI allowed for some insight into this area. The results found that 53% (nine musicians) wore hearing protection when listening to or playing music at high levels. However, the NESI also revealed that participants only used hearing protection 50% of time. Although the NESI did not directly assess the acceptability of hearing protection, some individuals expressed that the reason for low usage rates was due to sound quality and enjoyment (see also [38]). These results reflect those found by several other groups investigating hearing protection use in musicians [5–9]. As previously stated, these findings were not a direct aim of the study and as the musician group was made up of only 17 participants, the results should be interpreted cautiously.

Future research could focus on the underlying mechanisms that contribute to the differences in preferred music-listening level between musicians and non-musicians. Previous research in this area [11, 12] has found that loud music (above 90 dB SPL) comprising low frequencies between 100 to 500 Hz can activate the vestibular system and elicit a 'hedonic response'. The 'hedonic response' refers to the satisfactory feelings that result from this response. Previous research has highlighted that brain anatomy differs between musicians and non-musicians [25], therefore it could be hypothesized that central vestibular activity differs too. Therefore, it is conceivable that one of the underlying mechanisms is that musicians have less central vestibular activity and thus require higher sound-level stimulus in order for their vestibular system to activate and trigger the 'hedonic response'. Alternatively, musicians might have a more sensitive vestibular system (i.e. lower activation threshold or larger response) and thus be able to activate it and trigger the 'hedonic response' more easily than non-musicians. Consequently, musicians might experience more enjoyment from listening to music at high sound levels than non-musicians who might be unable to trigger their 'hedonic response' even with high-level sounds. In order to investigate such hypotheses, an assessment of the vestibular system (both central and peripheral) would need to be carried out for both groups. This could be achieved using clinically recognized methods such as magnetic resonance imaging and vestibular evoked myogenic potential [39]. Such research would enable insight into a possible underlying mechanism for the difference in preferred music-listening level, which in turn could provide invaluable information to be used to best protect individuals from NIHL. However, it is possible that there is no anatomical underlying mechanism for the difference in preferred music-listening level. Musicians may listen to music at higher levels in order to hear the more nuanced musical aspects or may simply enjoy music more so than non-musicians and are therefore happy to listen to music at a higher sound level.

## Limitations

In order to achieve a statistical power of 0.80 ($p < 0.05$) and observe a medium effect size a minimum of 128 participants was required (sample size estimated using G*Power, [40, 41]). However, we tested a smaller sample size, i.e., 34 participants. Despite this, the statistical analysis revealed the difference between musicians and non-musicians to be a Cohen's d of 1.14, which is a large effect size.

Additionally, It could be considered that the self-report measure of noise exposure has imprecisions and errors intrinsic to the recall approach (see [42], Section 1.3.10). Individuals who are more concerned about hearing loss may be more likely to overestimate the amount of noise exposure that they have. Conversely, those who are less concerned may underestimate.

## Conclusion

The main findings from the study are as follows:

- Musicians prefer to listen to music at a higher level than non-musicians. There was a 7-dB difference in preferred level between the two groups.

- Participants prefer listening to music they like at a higher level than music they disliked.

- Lifetime noise exposure correlates positively with preferred music-listening level.

Although more research is needed to better understand the underlying mechanisms, results from this study show that musical training and preference has some effect on preferred music-listening level. Additionally, sounds that are not deemed as noise can be tolerated and enjoyed at higher sound levels. These findings may have implications for the use of hearing protection to prevent NIHL, which may be particularly relevant for musicians with high levels of sound exposure.

## Supporting information

**S1 Table. Levels of the music samples in dB(A) when played through the audiometer.**
(DOCX)

**S2 Table. Pure tone audiometry.**
(XLSX)

**S3 Table. Preferred music-listening level data.**
(XLSX)

## Acknowledgments

We would like to thank Chris Plack and Sam Couth for their valuable comments on an earlier version of this manuscript. We are thankful to Antonia Marsden for advice with multiple regression models.

## Author Contributions

**Conceptualization:** Antonia Olivia Dolan, Karolina Kluk.

**Formal analysis:** Antonia Olivia Dolan, Emanuele Perugia.

**Funding acquisition:** Karolina Kluk.

**Investigation:** Antonia Olivia Dolan.

**Methodology:** Antonia Olivia Dolan.

**Project administration:** Karolina Kluk.

**Supervision:** Karolina Kluk.

**Visualization:** Emanuele Perugia.

**Writing – original draft:** Antonia Olivia Dolan, Karolina Kluk.

**Writing – review & editing:** Emanuele Perugia, Karolina Kluk.

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
