## [Decision Letter · Decision Letter 0]

2 May 2022

PONE-D-21-32770Preferred Music-Listening Level in Musicians and Non-MusiciansPLOS ONE

Dear Dr. Kluk,

Thank you for submitting your manuscript to PLOS ONE. Our apologies that the review of your paper has taken so long and you will see that we were only able to secure one reviewer.  After careful consideration, we feel that it has merit but does not fully meet PLOS ONE’s publication criteria as it currently stands. Therefore, we invite you to submit a revised version of the manuscript that addresses the points raised during the review process. The independent reviewer has a number of comments about the methodology, especially the NESI procedure and the sampling approach.  As editor, I concur with this and also have some additional comments which are attached including some reflections on the discussion which you should consider.

We look forward to receiving your revised manuscript.

Kind regards,

Peter Thorne, CNZM PhD

Academic Editor

PLOS ONE

Journal Requirements:

"* AOD, EP, KK

NIHR (National Institute for Health Research) Manchester Biomedical Research Centre

* EP, KK

MRC (Medical Research Council), UK (MR/L003589/1)"

Additional Editor Comments (if provided):

This is an interesting and overall well-written paper that looks at the preferred music listening levels in musicians and non-musicians. There are a number of areas which could be addressed to improve tje manuscript.

Line 91: More detail is needed on the characteristics of the participants, especially the types of gigs/concerts and instruments used by the musicians group. This could be an additional supplementary table

Line 143: Explain why the statistical analysis of the audiogram relates solely to 8kHz, given they have overlapping SD, or analyse the audiograms across the range of frequencies.

Line 170: It would be useful to have more descriptions about the NESI approach and how it was undertaken. Details of the type of noise exposure for all the participants would be of interest given that potentially the main noise exposure for the musicians was the concerts and gigs. It would help to understand the relationship of the effect of NESI on preferred listening levels and musicianship (line 261) and the possibility of hearing damage as a cause of the relationship in the discussion.

What was the extent of hearing protection use in these participants given the focus on the importance of hearing protection in the paper. This appears in the discussion only as a comment.

Line 273. provide a reference for the activation of the IPL in musicians and the role of the IPL in perception.

The structure of the discussion implies that the 3 aims are independent. The possibility of hearing damage from excessive sound exposure from continual loud music could be the key factor and needs to be discussed more relative to the type of exposures and the segmentation across the lifespan. This is mentioned, and the possibility of synaptopathy is a good point, but generally it is glossed over in favour of other theories, including vestibular activation for which no data is presented. Unfortunately because of the lack of testing in the extended high frequencies and lack of any other audiometry testing, such as speech reception in noise it is not possible to rule out a hearing deficit as a cause of the relationship of musicianship and preferred listening levels. The possibility of hearing damage as a significant contributing factor needs to be more strongly emphasized in the conclusions along with the importance of future research that investigates the integrity of high frequency hearing and hearing performance in musicians when investigating music listening levels.

Reviewers' comments:

Reviewer's Responses to Questions

**Comments to the Author**

1. Is the manuscript technically sound, and do the data support the conclusions?

Reviewer #1: Partly

2. Has the statistical analysis been performed appropriately and rigorously? 

Reviewer #1: Yes

3. Have the authors made all data underlying the findings in their manuscript fully available?

Reviewer #1: Yes

4. Is the manuscript presented in an intelligible fashion and written in standard English?

Reviewer #1: Yes

5. Review Comments to the Author

Reviewer #1: Thank you for the opportunity to review this manuscript. Please find below my comments for your consideration.

ABSTRACT:

- The last sentence of the abstract concludes that, '...use of hearing protection is vital for their hearing health and career longevity'. How is hearing protection related to music listening levels - which is the focus of your study? Your three main findings relate to listening levels specifically. Perhaps, you would want to consider a concluding remark related to safe listening and hearing conservation (which will include elements of hearing protection for musicians when playing instruments).

MATERIALS AND METHODS:

- Participants - how were the participants identified and recruited? What was the sampling/recruitment strategy?

- Preferred music level - 'Following the hearing screening, participants were presented with six music samples and one environmental sound...'. What were the six music samples and environmental sound? The bird sound is first mentioned in the 'results' section. It needs to be introduced in the methods.

- What was the duration of the sample sounds if they were not stopped?

- 'This process was repeated three times and an average value was determined for each music sample'. Was there a break between the three times or was it simultaneous?

- Noise exposure - How long does it take to conduct the NESI interview?

- The original Guest paper on NESI, ' The methods by which noise exposure data are obtained and combined fall into seven basic categories: (a) identification of exposure activities, (b) segmentation of the lifespan, (c) estimation of exposure duration, (d) estimation of exposure level, (e) consideration of hearing protection, (f) quantification of firearm noise exposure, and (g) calculation of noise exposure units'. Did you do this in your research when using NESI? This section needs greater detail. For example, were participants given examples of different sources of noise exposure? Was there segmentation of lifespan?

- Analysis - How was the NESI score calculated?

DISCUSSION: If the above questions related to NESI are clarified, there can be a better judgement of the discussion of noise exposure on page 13. Given that you categorized noise exposure into occupational, recreational and firearm use, my assumption is that musicians will have high occupational and recreational noise exposure purely by nature of their work and interest. That would mean that the comparable group should have been recruited from similar occupational groups where there is noise exposure as part of their daily work.

LIMITATIONS: My suggestion is to remove 'time constraints' as a reason for the smaller sample size. Instead just state that you had a smaller sample than what was planned.

- Page 16, ' Additionally, it could be considered that the use of a subjective self-report measure of noise exposure (see [35], Section 1.3.10)'. I am not sure what this sentence is about?

6. PLOS authors have the option to publish the peer review history of their article (what does this mean?). If published, this will include your full peer review and any attached files.

Reviewer #1: No

---

## [Author Response · Author response to Decision Letter 0]

20 Oct 2022

We thank the reviewer and the academic editor for their comments. In our revised manuscript we have attempted to address all points raised by the Reviewers.

To make it easy for the reviewers and editor to see how we have responded to their comments, we have separated and numbered their comments. We have used italicised font and blue for the reviewer’s comments.

Editor: 

This is an interesting and overall well-written paper that looks at the preferred music listening levels in musicians and non-musicians. There are a number of areas which could be addressed to improve the manuscript.

Thank you for your comments. We have attempted to improve the focus of the manuscript.

Detailed comments:

 Line 91: More detail is needed on the characteristics of the participants, especially the types of gigs/concerts and instruments used by the musicians group. This could be an additional supplementary table

 Unfortunately, we did not collect individual information about the gigs/concerts and instruments used by our musicians. We added the following in the text: 

“The musicians ranged from guitarist/bassist, drummer to saxophonist. They played in rock and jazz bands.”

 Line 143: Explain why the statistical analysis of the audiogram relates solely to 8kHz, given they have overlapping SD, or analyse the audiograms across the range of frequencies.

 Thank you to point it out. We ran the analysis solely at 8 kHz because it was the only frequency with some noticeable difference between musician and non-musician. However, following your advice we have now included all frequencies:

“Using the average over the two ears, musicians and non-musicians had similar average thresholds from 250 to 8000 Hz (W = 4669.5, p = 0.202) musicians had the median = 2.5 dB HL (SD = 5.931), whereas the non-musicians had median = 5 dB HL (SD = 6.151) .”

 Line 170: It would be useful to have more descriptions about the NESI approach and how it was undertaken. Details of the type of noise exposure for all the participants would be of interest given that potentially the main noise exposure for the musicians was the concerts and gigs. It would help to understand the relationship of the effect of NESI on preferred listening levels and musicianship (line 261) and the possibility of hearing damage as a cause of the relationship in the discussion.

 We added the formula and its description to estimate the NESI score

“ For a given activity, the cumulative units of noise exposure are calculated as:

U=(Y*W*D*H)/2080*[P*10^((L-A-90)/10)+(1-p)*10^((L-90)/10) ]

Where U is units of noise exposure (in linear units), Y is years of exposure, W is weeks per year of exposure, D is days per week of exposure, H is hours per day of exposure, 2080 is the number of hours in a working year, L is the estimated sound level in dBA, A is the attenuation of hearing protection (dB), and P is the proportion of time that hearing protection was worn (0 to 1). The values of U were calculated for each activity and then summed”

 We agree that details of the type of noise exposure for all the participants is of interest and we added the following in the results section: 

“Overall, the NESI scores were between 1.58 and 126.78. On average 81% of the overall NESI score was driven by recreational activities, whilst the 19% was due to occupational activities. All 34 participants took part in some ‘noisy’ recreational activities. However, 24 participants, of which 15 were musicians, reported ‘noisy’ occupational activities. Twelve participants (7 musicians) used firearms but with a decimal NESI score.”

 What was the extent of hearing protection use in these participants given the focus on the importance of hearing protection in the paper. This appears in the discussion only as a comment.

 Added: 

“The hearing protection used ranged from Custom in-the-ear (IE) Plugs (ER bang series or similar alternatives with filters used according to instrument played) to standard foam ear plugs.”

 Line 273. provide a reference for the activation of the IPL in musicians and the role of the IPL in perception.

 Reference provided:

Liu et al. (2018) and Limb (2006) for activation of the IPL

Liu et al. (2018) and Platel (1997) for the role of IPL in perception

 The structure of the discussion implies that the 3 aims are independent. The possibility of hearing damage from excessive sound exposure from continual loud music could be the key factor and needs to be discussed more relative to the type of exposures and the segmentation across the lifespan. This is mentioned, and the possibility of synaptopathy is a good point, but generally it is glossed over in favour of other theories, including vestibular activation for which no data is presented. Unfortunately, because of the lack of testing in the extended high frequencies and lack of any other audiometry testing, such as speech reception in noise it is not possible to rule out a hearing deficit as a cause of the relationship of musicianship and preferred listening levels. The possibility of hearing damage as a significant contributing factor needs to be more strongly emphasized in the conclusions along with the importance of future research that investigates the integrity of high frequency hearing and hearing performance in musicians when investigating music listening levels.

 We emphasised the lack of extended high frequencies and other audiometry testing:

“However, we cannot rule out any (sub-clinical) hearing damage in musicians as extended high-frequency audiometry or speech-in-noise tests were not performed. Further studies should include these measures as they may provide further understanding for why musicians prefer to listen to music at higher levels than non-musicians.”

 We discussed further the possibility of synaptopathy adding:

“Couth et al. (2020) investigated the perceptual consequences of NICS in musicians using an extensive test battery. Although musicians reported poorer hearing in noise abilities than non-musicians, Couth et al. (2020) did not observe any significant difference between musicians and non-musicians, or between high and low-noise exposure groups, in any of the tests in the test battery. It is possible that this lack of difference in performance between musicians and non- musicians was due t0 participants with more musical training being better at temporal processing tasks than naive participants (Yeend et al 2017, Perugia et al 2021, Prendergast et al. 2017). “

Reviewer #1: 

ABSTRACT:

 The last sentence of the abstract concludes that, '...use of hearing protection is vital for their hearing health and career longevity'. How is hearing protection related to music listening levels - which is the focus of your study? Your three main findings relate to listening levels specifically. Perhaps, you would want to consider a concluding remark related to safe listening and hearing conservation (which will include elements of hearing protection for musicians when playing instruments).

 Thank you. We changed this into: 

“As such, musicians in particular would benefit from simple changes in lifestyle and listening habits, including increased awareness of the risks of higher listening levels, as well as the use of hearing protection.”

MATERIALS AND METHODS:

 Participants - how were the participants identified and recruited? What was the sampling/recruitment strategy?

 Added: 

“Participants were recruited using flyers advertising the research,et

around the University of Manchester campus and at the ‘Futureworks’ campus (i.e., higher education setting for music performance and audio engineering)”

 Preferred music level - 'Following the hearing screening, participants were presented with six music samples and one environmental sound...'. What were the six music samples and environmental sound? The bird sound is first mentioned in the 'results' section. It needs to be introduced in the methods.

 Added: 

“The six music samples were: 1) Whole Lotta Love by Led Zeppelin, 2) Heartbeats by Jose Gonzales, 2) Crazy In Love by Beyonce, 3) Sad But True by Metallica, 4) Virtual Insanity by Jamiroquai, 6) Symphony No. 5 in C Minor, Op. 67: I. Allegro con brio by Ludwig Van Beethoven et al. Whilst, the environment sound was birds song, which was a sample taken from a high-quality recording of birds signing.”

 What was the duration of the sample sounds if they were not stopped?

 Changed: 

“The music/environmental samples lasted 60 seconds and varied in genre”

 'This process was repeated three times and an average value was determined for each music sample'. Was there a break between the three times or was it simultaneous?

 There was only a little break between the three repetitions due to reset of the playback, otherwise it was continuous. Added: 

“The repetitions were performed consecutively”

 Noise exposure - How long does it take to conduct the NESI interview?

 Added: 

“The interviews took about 20 minutes to complete.”

 The original Guest paper on NESI, ' The methods by which noise exposure data are obtained and combined fall into seven basic categories: (a) identification of exposure activities, (b) segmentation of the lifespan, (c) estimation of exposure duration, (d) estimation of exposure level, (e) consideration of hearing protection, (f) quantification of firearm noise exposure, and (g) calculation of noise exposure units'. Did you do this in your research when using NESI? This section needs greater detail. For example, were participants given examples of different sources of noise exposure? Was there segmentation of lifespan?

 Yes, we did use the seven categories and gave example to the participants. We added:

“In the interviews, the supplementary materials provided by Guest et al. (2018), such as guidance (with examples and tables) and spreadsheet, to calculate the total units of lifetime noise exposure.”

 We added in the results also: 

“Overall, the NESI score was between 1.577 and 126.779. The 80.46% of the NESI score was due to recreational activities, whilst the 19.01% was due to occupational activities. All 34 participants took part in some ‘noisy’ recreational activities. However, 24 participants, of which 15 musicians, identified ‘noisy’ occupational activities. Twelve participants (7 musicians) used firearms but with a decimal NESI score.”

 Analysis - How was the NESI score calculated?

 We added the formula and its description to estimate the NESI score

“ For a given activity, the cumulative units of noise exposure are calculated as:

U=(Y*W*D*H)/2080*[P*10^((L-A-90)/10)+(1-p)*10^((L-90)/10) ]

Where U is units of noise exposure (in linear units), Y is years of exposure, W is weeks per year of exposure, D is days per week of exposure, H is hours per day of exposure, 2080 is the number of hours in a working year, L is the estimated sound level in dBA, A is the attenuation of hearing protection (dB), and P is the proportion of time that hearing protection was worn (0 to 1). The values of U were calculated for each activity and then summed.” 

DISCUSSION:

 If the above questions related to NESI are clarified, there can be a better judgement of the discussion of noise exposure on page 13. Given that you categorized noise exposure into occupational, recreational and firearm use, my assumption is that musicians will have high occupational and recreational noise exposure purely by nature of their work and interest. That would mean that the comparable group should have been recruited from similar occupational groups where there is noise exposure as part of their daily work.

 Thank you we added:

“The recruitment was done using flyers around the campus and at the ‘Futureworks’ campus, so the greater noise exposure in musicians than non-musicians may be due to the intrinsic musicians’ interests and works. For instance, ‘noisy’ occupational activities were reported by 15 out of 17 musicians but only by 9 out of 17 non-musicians.”

LIMITATIONS:

 My suggestion is to remove 'time constraints' as a reason for the smaller sample size. Instead just state that you had a smaller sample than what was planned.

 Thank you for the suggestion. We changed accordingly:

“However, we tested a smaller sample size, i.e., 34 participants.” 

 Page 16, ' Additionally, it could be considered that the use of a subjective self-report measure of noise exposure (see [35], Section 1.3.10)'. I am not sure what this sentence is about?

 We do agree that the original wording was clumsy. We changed:

“It could be considered that the self-report measure of noise exposure has imprecisions and errors intrinsic to the recall approach”.

---

## [Decision Letter · Decision Letter 1]

25 Nov 2022

Preferred Music-Listening Level in Musicians and Non-Musicians

PONE-D-21-32770R1

Dear Dr. Kluk,

Thank you for your re-submitted manuscript and for addressing the issues raised during the review. We’re pleased to inform you that your manuscript has been judged scientifically suitable for publication and will be formally accepted for publication once it meets all outstanding technical requirements.

Kind regards,

Peter Thorne, 

Academic Editor

PLOS ONE

Additional Editor Comments (optional):

The issues and comments are all suitably addressed. Thank you.

Reviewers' comments:

Reviewer's Responses to Questions

**Comments to the Author**

1. If the authors have adequately addressed your comments raised in a previous round of review and you feel that this manuscript is now acceptable for publication, you may indicate that here to bypass the “Comments to the Author” section, enter your conflict of interest statement in the “Confidential to Editor” section, and submit your "Accept" recommendation.

Reviewer #1: All comments have been addressed

2. Is the manuscript technically sound, and do the data support the conclusions?

Reviewer #1: Yes

3. Has the statistical analysis been performed appropriately and rigorously? 

Reviewer #1: Yes

4. Have the authors made all data underlying the findings in their manuscript fully available?

Reviewer #1: Yes

5. Is the manuscript presented in an intelligible fashion and written in standard English?

Reviewer #1: Yes

6. Review Comments to the Author

Reviewer #1: Thank you for the response to the review. These were satisfactorily addressed. You may consider adding an anonymized dataset as supplementary material.

7. PLOS authors have the option to publish the peer review history of their article (what does this mean?). If published, this will include your full peer review and any attached files.

Reviewer #1: No

---

## [Editor Report · Acceptance letter]

12 Dec 2022

PONE-D-21-32770R1 

Preferred Music-Listening Level in Musicians and Non-Musicians 

Dear Dr. Kluk:

I'm pleased to inform you that your manuscript has been deemed suitable for publication in PLOS ONE. Congratulations! Your manuscript is now with our production department. 

Kind regards, 

on behalf of

Dr. Peter Rowland Thorne 

Academic Editor

PLOS ONE